Regulation of KCa2.3 and endothelium-dependent hyperpolarization (EDH) in the rat middle cerebral artery: the role of lipoxygenase metabolites and isoprostanes

Gauthier Kathryn M. 1
Campbell William B. 1
McNeish Alister J. 2 a.mcneish@reading.ac.uk
1 Department of Pharmacology and Toxicology, Medical College of Wisconsin , Milwaukee, WI , USA
2 Reading School of Pharmacy, University of Reading , Reading, Berkshire , UK
Silva Jerson
Electronic publication date: 2014 Jun 10
Publication date: 2014
Volume: 2
Electronic Location ID: e414
Received 2014 Feb 28; Accepted 2014 May 15
Copyright: © 2014 Gauthier et al.
Copyright year: 2014
Copyright holder: Gauthier et al.
License: This is an open access article distributed under the terms of the Creative Commons Attribution License, which permits unrestricted use, distribution, reproduction and adaptation in any medium and for any purpose provided that it is properly attributed. For attribution, the original author(s), title, publication source (PeerJ) and either DOI or URL of the article must be cited.
License URL: https://creativecommons.org/licenses/by/4.0/

Keywords: Endothelium, Lipoxygensase, Isoprostane, Calcium-activated potassium channels, Nitric oxide, EDH, EDHF, Cerebral artery, Vasodilation, Arachidonic acid

Funding: British Heart Foundation FS/06/076/21988 PG/11/93/29143 National Heart, Lung and Blood Institute HL-103673 HL-37981 This work was supported by the British Heart Foundation (FS/06/076/21988 and PG/11/93/29143) and the National Heart, Lung and Blood Institute (HL-103673 and HL-37981). The funders had no role in study design, data collection and analysis, decision to publish, or preparation of the manuscript.

==============================
Background and Purpose. In rat middle cerebral arteries, endothelium-dependent hyperpolarization (EDH) is mediated by activation of calcium-activated potassium (KCa) channels specifically KCa2.3 and KCa3.1. Lipoxygenase (LOX) products function as endothelium-derived hyperpolarizing factors (EDHFs) in rabbit arteries by stimulating KCa2.3. We investigated if LOX products contribute to EDH in rat cerebral arteries.

Methods. Arachidonic acid (AA) metabolites produced in middle cerebral arteries were measured using HPLC and LC/MS. Vascular tension and membrane potential responses to SLIGRL were simultaneously recorded using wire myography and intracellular microelectrodes.

Results. SLIGRL, an agonist at PAR2 receptors, caused EDH that was inhibited by a combination of KCa2.3 and KCa3.1 blockade. Non-selective LOX-inhibition reduced EDH, whereas inhibition of 12-LOX had no effect. Soluble epoxide hydrolase (sEH) inhibition enhanced the KCa2.3 component of EDH. Following NO synthase (NOS) inhibition, the KCa2.3 component of EDH was absent. Using HPLC, middle cerebral arteries metabolized 14C-AA to 15- and 12-LOX products under control conditions. With NOS inhibition, there was little change in LOX metabolites, but increased F-type isoprostanes. 8-iso-PGF2α inhibited the KCa2.3 component of EDH.

Conclusions. LOX metabolites mediate EDH in rat middle cerebral arteries. Inhibition of sEH increases the KCa2.3 component of EDH. Following NOS inhibition, loss of KCa2.3 function is independent of changes in LOX production or sEH inhibition but due to increased isoprostane production and subsequent stimulation of TP receptors. These findings have important implications in diseases associated with loss of NO signaling such as stroke; where inhibition of sEH and/or isoprostane formation may of benefit.

Introduction

Endothelium dependent hyperpolarization (EDH) is an extremely important vasodilator pathway, particularly in resistance size arteries, and has a major role in regulation of blood pressure; as such the EDH pathway has become intensively studied and targeting this pathway could result in new classes of antihypertensive and possibly antithrombotic therapies (Kohler, Kaistha & Wulff, 2010; Krötz et al., 2010). The exact nature of EDH is still unresolved but involves several different mechanisms that include release of diffusible factors (endothelium derived hyperpolarizing factors; EDHFs) (Mombouli et al., 1996), direct myo-endothelial cell coupling via gap junctions (Sandow et al., 2002) and increases in extracellular potassium concentration in the myo-endothelial space (Edwards et al., 1998) that all result in smooth muscle cell hyperpolarization. The relative contribution of each mechanism varies between vascular beds and species (Edwards, Feletou & Weston, 2010).

Despite heterogeneity in the pathway, a feature of EDH responses in most vascular beds is the critical role of endothelial cell calcium-activated potassium channels (KCa). Blockade of both KCa2.3 (SKCa) and KCa3.1 (IKCa) is required to fully inhibit EDH responses (Edwards, Feletou & Weston, 2010). In contrast, in the rat middle cerebral artery (MCA), inhibition of KCa3.1 channels alone is sufficient to fully block the EDH response obtained but only if NO synthase (NOS) is inhibited (Marrelli, Eckmann & Hunte, 2003; McNeish, Dora & Garland, 2005). KCa2.3 channels are expressed in the endothelium of MCA (Cipolla et al., 2009; McNeish et al., 2006) and contribute to EDH of smooth muscle cells (SMC) in vessels that are able to synthesize NO (McNeish et al., 2006). KCa2.3 channel function is lost following inhibition of NOS (McNeish et al., 2006), but can be restored upon inhibition of the thromboxane prostanoid (TP) receptor and associated subsequent Rho kinase mediated signalling (McNeish & Garland, 2007; McNeish et al., 2012).

In the MCA, EDH responses are also attenuated by inhibition of PLA2 suggesting a role for a metabolite of arachidonic acid (AA) (Marrelli et al., 2007; McNeish & Garland, 2007; You, Marrelli & Bryan, 2002). A similar inhibition of EDH by PLA2 inhibitors has also been reported in peripheral arterial beds including mesenteric and coronary arteries (Hecker et al., 1994; Hutcheson et al., 1999). AA metabolism is complex and produces a large number of biologically active metabolites that affect vascular tone (Campbell & Falck, 2007). AA metabolites produced by the cyclooxygenase (COX), epoxygenase and lipoxygenase (LOX) pathways modulate KCa. The 12- and 15-LOX pathways of AA metabolism have been implicated in the EDH response (Campbell et al., 1996; Chawengsub, Gauthier & Campbell, 2009; Nithipatikom et al., 2001a). Metabolites of 15-LOX, 15-hydroxy-11,12-epoxyeicosatrienoic acid (HEETA) and its soluble epoxide hydrolase (sEH) metabolite, 11,12,15-trihydroxyeicosatrienoic acid (THETA), have been characterized as EDHFs in rabbit arteries where they stimulate KCa2.3 channels (Campbell & Gauthier, 2013). Rats express a LOX that has both 12- and 15-LOX activity (Watanabe et al., 1993). Therefore, it is possible that LOX metabolites acting on KCa2.3 could also contribute EDH in the rat MCA.

NO can potentially interact with the synthesis of many AA metabolites by binding to the heme group of these enzymes (Minamiyama et al., 1997). Therefore, TP receptor-mediated inhibition of KCa2.3 following inhibition of NOS (McNeish & Garland, 2007) may be due to altered synthesis of AA metabolites such as thromboxane A2 (TxA2) or isoprostanes that stimulate the TP receptor. Alternately following inhibition of NOS, there may be a change in production of AA metabolites that normally stimulate KCa2.3.

As changes in AA metabolism following inhibition of NOS could potentially affect KCa2.3 function in the MCA, these studies were designed to identify the AA metabolites and determine if these metabolites contribute to EDH in the MCA. Second, we investigated the effect of NOS inhibition on AA metabolism and if any change in metabolite production contributes to the loss of KCa2.3 function.

Materials and Methods

Male Wistar or Sprague Dawley rats (200–300 g, circa 8–10 weeks old) were humanely killed in accordance with local ethics committee guidelines. The brain was removed and immediately placed in ice-cold Krebs’ solution. Segments of the middle cerebral artery (∼2 mm long) were dissected and stored in ice-cold Krebs’ for use (usually within 30 min but if first vessel failed to meet the exclusion criteria detailed below a second vessel would be mounted this was never >4 h), with similar size vessels used in all experimental groups.

Simultaneous measurement of membrane potential and tension

Smooth muscle membrane potential was measured simultaneously with vasomotor responses using isometric recording technique. Segments of MCA (internal diameter ∼150 µm) were mounted in a Mulvany–Halpern myograph (model 400A; Danish Myotechnology) in Krebs’ solution containing (mM): NaCl, 118.0, NaCO3, 24; KCl, 3.6; MgSO4⋅7H2O, 1.2; glucose, 11.0; CaCl2, 2.5; gassed with 21% O2, 5% CO2, balance N2 and maintained at 37 °C. After equilibration for 20 min, vessels were tensioned to 1–1.5 mN (approximates wall tension at 60 mmHg). Smooth muscle tension was recorded with an isometric pressure transducer and Powerlab software (ADI, Australia). Vessel viability was assessed by adding exogenous K+ (15–55 mM, total K+ concentration); only vessels developing tension of ≥3 mN were used. Endothelial cell viability was assessed by the ability of an agonist of protease activated receptor 2, SLIGRL (serine, leucine, isoleucine, glycine, arginine, leucine; 20 µM) to relax spontaneous tone and to hyperpolarize the smooth muscle cell membrane by >15 mV. SLIGRL was chosen to stimulate the endothelium as more commonly used agonists such as ACh are unable to evoke an EDH response in the MCA and purinergic agonists that can stimulate EDH responses evoke constriction via smooth muscle cell action.

All endothelium-dependent responses to SLIGRL (20 µM) were obtained in resting vessels with no vasoconstrictor stimulus other than spontaneous tone unless stated otherwise. EDH of smooth muscle cells was assessed in the presence of the KCa channel blockers, apamin (KCa2.3, 50 nM), TRAM-34 (KCa3.1, 1 µM) and iberiotoxin (KCa1.1, 100 nM). The effect of KCa blockers on EDH responses was also assessed after addition of three structurally distinct non-selective inhibitors of LOX (ebselen, nordihydroguaiaretic acid (NGDA), both 1 µM, and PD146176, 5 µM), soluble epoxide hydrolase (sEH) inhibitor tAUCB (10 µM) and isoprostane isoPGF2α (1 µM). Isolated EDH responses were obtained in the presence of the NO synthase inhibitor L-NAME (100 µM). In these experiments, the effect of the KCa channel blockers was assessed on EDH responses induced by SLIGRL (20 µM) in the presence of tAUCB or tempol (100 µM). Papaverine (150 µM) was added at the end of each experiment to assess overall tone and all relaxations are expressed as a percentage of this maximal relaxation. All drugs were allowed to equilibrate for at least 20 min before EDH responses were stimulated. In most experiments, smooth muscle membrane potential (Em) and tension were measured simultaneously as previously described, using glass microelectrodes (filled with 2 M KCl; tip resistance, 80–120 MΩ) to measure Em (Garland & McPherson, 1992). In experiments where vessels were able to synthesise NO we only reported EDH of the smooth muscle cell layer as NO was able to evoke relaxation even if EDH was blocked.

Measurement of metabolites of 14C-labelled AA

Rat cerebral arteries were dissected, cleaned, cut into 2–3 mm rings, and incubated at 37 °C with indomethacin (10 µM) and vehicle or indomethacin and L-NAME (100 µM) in 2 ml of N-2-hydroxyethylpiperazine-N′-2-ethanesulfonic acid (HEPES) buffer (mM): 10 HEPES, 150 NaCl, 5 KCl, 2 CaCl2, 1 MgCl2, 6 glucose, pH 7.4. After 10 min, [14C]-AA (0.5 µCi, 10−7 M) was added, incubation was continued for 5 min, and then A23187 (10 µM) was added. After 15 min, the reaction was stopped with ethanol (15% final concentration). The incubation buffer was removed and extracted using Bond Elute octadecylsilyl columns as previously described (Pfister et al., 1998). The extracts of the media were analyzed by reverse phase high-pressure liquid chromatography (HPLC) using a Nucleosil C-18 (5μ, 4.6 × 250 mm) column (Pfister et al., 1998). Solvent system 1 was used to resolve lipoxygenase and epoxygenase metabolites and consisted of a 40 min linear gradient (flow rate = 1 ml/min) from 50% solvent B (acetonitrile with 0.1% glacial acetic acid) in solvent A (deionized water) to 100% solvent B. Column effluent was collected in 0.2 ml fractions, and the radioactivity was determined. In a separate series of incubations, arterial rings were incubated with vehicle or L-NAME (100 µM) as described above. Following extraction of the media, prostaglandins and isoprostanes were resolved using a Nucleosil C-18 column using solvent system 2. This solvent system consisted of an 40 min isocratic elution with 31% solvent B (acetonitrile) in solvent A (water containing 0.025 M phosphoric acid) followed by a 20 min linear gradient to 100% B and a 10 min elution with 100% B. The column eluate was collected in 0.5 ml fractions and radioactivity determined. Total radioactivity from a sample was normalized to tissue weight.

The extracts were analyzed using liquid chromatography–mass spectrometry (LC/MS) with an Agilent Model 1100SL mass selective detector and electrospray ionization (Nithipatikom et al., 2001b; Nithipatikom et al., 2003). Metabolites were resolved using a Kromasil C-18 (5 µm, 2 × 250 mm) column. Solvent system 3 consisted of a 10 min linear gradient from 15% solvent B (acetonitrile) in solvent A (water containing 0.005% glacial acetic acid) to 60% solvent B in A, a 10 min linear gradient to 80% solvent B in A followed by 5 min gradient to 100% B. The flow rate was 0.2 ml/min. The drying gas flow was 12 l/min. The nebulizer pressure was 35 psig, vaporizer temperature was 325 °C and desolvation temperature 350 °C. The capillary voltage was 3 kV and fragmentor voltage was 120 V. Detection was in the negative ion mode. For isoprostanes, the eluate was monitored at m/z 351 (M-H) for PGE2-type isoprostanes and m/z 353 for PGF2-type isoprostanes. Comparisons were made with the elution times of known isoprostane standards.

Data analysis and statistical procedures

Results are expressed as the mean ± s.e.m. of n animals. Tension values are given in mN (always per 2 mm segment) and Em as mV. Vasodilatation is expressed as percentage reduction of the total vascular tone (spontaneous tone plus vasoconstrictor response), quantified by relaxation with papaverine (150 µM). Graphs were drawn and comparisons made using one-way ANOVA with Tukeys’ post-test or Students’ t-test, as appropriate (Prism, GraphPad). P ≤ 0.05 was considered significant.

Drugs, chemicals, reagents and other materials

Exogenous K+ was added as an isotonic physiological salt solution in which all the NaCl was replaced with an equivalent amount of KCl. Concentrations of K+ used are expressed as final bath concentration. Ebselen (2-Phenyl-1,2-benzisoselenazol-3(2H)-one), L-NAME (NG-nitro-L-arginine methyl ester), Nordihydroguaiaretic acid (NDGA), papaverine HCl, PD 146176 (6,11-Dihydro[1]benzothiopyrano[4,3-b]indole), tempol (4-Hydroxy-2,2,6,6-tetramethylpiperidine 1-oxyl), TRAM-34 (1-[(2-Chlorophenyl)diphenylmethyl]-1H-pyrazole) and U46619 (9,11-Dideoxy-11α,9α-epoxymethanoprostaglandin F2α) were all obtained from Sigma (Poole, U.K.). Apamin and iberotoxin, from Latoxan (Valence, France). SLIGRL-NH2 (serine, leucine, isoleucine, glycine, arginine, leucine) from Auspep (Parkville, Australia). 8-iso PGF2α from Cayman-Europe (Tallinn, Estonia). tAUCB (trans-4-[4-(3-Adamantan-1-yl-ureido)-cyclohexyloxy]-benzoic acid) was a generous gift from Professor Bruce D. Hammock (University of California, Davis). All stock solutions (100 mM) were prepared in dimethylsulfoxide (DMSO) except L-NAME, apamin, iberiotoxin, papaverine and SLIGRL that were dissolved in 0.9% NaCl and tempol which was dissolved in ultrapure water. Vehicle controls were performed when necessary.

Results

Effect of LOX and sEH inhibition on EDH

In our previous studies, control arteries that had not been treated with a NOS inhibitor demonstrate endothelium-dependent hyperpolarization (EDH) that has components from KCa2.3, KCa3.1 and KCa1.1 channels (McNeish & Garland, 2007; McNeish et al., 2006). A similar pattern was seen in this study. Control EDH was significantly inhibited only by the combination of the KCa3.1 inhibitor TRAM-34 and the KCa2.3 inhibitor apamin; TRAM-34 alone had no significant inhibition alone (Fig. 1B); apamin alone was not assessed in this study but our previous studies demonstrate alone it does not have a significant effect on EDH under these conditions (McNeish & Garland, 2007; McNeish et al., 2006). The residual EDH was blocked by the inhibitor of KCa1.1 (BKCa), iberiotoxin. SLIGRL (20 µM) evoked EDH (−30.0 ± 5.5 mV, n = 4) that was significantly reduced by the LOX inhibitor nordihydroguaiaretic (NDGA, 1 µM; −10.7 ± 2.3 mV; n = 4, P < 0.05; Figs. 1A and 1C). This hyperpolarization was further reduced by TRAM-34 (−4.3 ± 2.4 mV, n = 5) but not by subsequent additions of apamin and iberiotoxin (Fig. 1B). NDGA also abolished spontaneous tone (1.9 ± 0.6 vs 0.2 ± 0.2 mN, pre- and post-NDGA, respectively, n = 4, P < 0.05). This relaxation was not associated with significant hyperpolarization (Em−44.6 ± 3.0 vs 51.9 ± 4.3 mV, pre- and post-NDGA, respectively, n = 4). A structurally distinct blocker of LOX, ebselen (10 µM), had similar effects. Ebselen significantly inhibited EDH (−28.8 ± 2.2 mV vs −15.6 ± 2.9 mV, pre- and post-ebselen, respectively; P < 0.05, n = 6; Fig. 1C). Hyperpolarization that was resistant to ebselen was not significantly reduced by TRAM-34 (Fig. 1C) or further addition of apamin but the subsequent addition of iberiotoxin reduced hyperpolarization further (Fig. 1D). Ebselen also reduced spontaneous tone (0.6 ± 0.0 mN vs 0.3 ± 0.1 mN, pre- and post-ebselen, respectively, n = 6, P < 0.05) that was not associated with significant hyperpolarization (Em−44.3 ± 1.0 vs −49.0 ± 3.3 mV, pre- and post-ebselen, respectively, n = 6). A third LOX inhibitor, PD146176 (5 µM), also reduced EDH (−27.5 ± 3.1 and −9.8 ± 2.6 mV pre- and post-PD146176 respectively, n = 5, P < 0.05). PD146176 also reduced spontaneous tone (1.5 ± 0.3 and 0.2 ± 0.1 mN pre- and post-PD146176 respectively, n = 5, P < 0.05) that was not associated with significant hyperpolarization (Em−52.1 ± 3.9 vs −58.7 ± 4.5 mV, pre- and post-PD146176, respectively, n = 5). The similar effect of these three structurally different LOX inhibitors to inhibit EDH and abolish the KCa2.3 component of this response indicates a role for LOX-derived AA metabolites in EDH.

Figure 1 Effect of lipoxygenase inhibition on endothelium-dependent hyperpolarization (EDH) evoked by SLIGRL (20 µM) in rat middle cerebral arteries that are able to synthesize NO.

(A) An original, representative trace showing control EDH (upper panel) and the effect of the non-selective lipoxygenase inhibitor NDGA (1 µM) on EDH (lower panel). Also shown are bar graphs showing (B) control EDH and the effect of lipoxygenase inhibitors NDGA (C) and ebselen (1 µM) (D) as well as the effect subsequent inhibition of KCa3.1 (TRAM-34) KCa2.1 (apamin) and KCa1.1 (iberiotoxin) on EDH. Control EDH was only inhibited following inhibition of KCa2.3 and KCa3.1 in the combined presence of apamin and TRAM-34, respectively. NDGA and ebselen significantly inhibited EDH, the residual hyperpolarization was inhibited by TRAM-34 alone, subsequent addition of apamin and IbTx had no significant additional effect. ∗P < 0.05 indicates a significant difference from control.

Previous studies indicate that the LOX-derived HEETA is metabolized by sEH to THETA (Chawengsub et al., 2008). Inhibition of sEH reduced HEETA metabolism and enhanced EDH to AA suggesting an important role for HEETA in mediating EDH. The inhibitor of sEH, tAUCB (10 µM) did not significantly effect EDH to SLIGRL in control vessels (−33.5 ± 5.4 vs 30.8 ± 6.0 mV pre- and post-tAUCB, respectively n = 5). Following inhibition of KCa3.1 and KCa1.1 with the combination TRAM-34 and iberiotoxin, the residual hyperpolarization to SLIGRL (−17.8 ± 2.8 mV, n = 5) is mediated solely by KCa2.3. Under these conditions, the KCa2.3-mediated hyperpolarization was significantly augmented by tAUCB (−29.2 ± 3.6, P < 0.05, n = 6, Figs. 2A and 2B). This augmented response is specific for KCa2.3 since addition of apamin significantly inhibited EDH (−7.0 ± 2.0 mV, P < 0.05, Figs. 2A and 2B). Therefore, an epoxide-containing LOX metabolite of AA contributes to the KCa2.3 component of EDH and is normally metabolized by sEH.

Figure 2 The effect of the sEH inhibitor tAUCB on the KCa2.3 component of endothelium-dependent hyperpolarization (EDH) in rat middle cerebral arteries able to synthesize NO.

(A) Original traces of KCa2.3-mediated hyperpolarization obtained in the presence of: the KCa3.1 inhibitor TRAM-34 and KCa1.1 inhibitor iberiotoxin (upper trace), the subsequent addition of tAUCB (middle trace) and the combination of tAUCB and the KCa2.3 inhibitor, apamin. (B) Bar graph showing control EDH, KCa2.3 dependent EDH in the presence and absence of tAUCB and further addition of apamin. The KCa2.3-dependent hyperpolarization was significantly smaller than control EDH and was potentiated by tAUCB this hyperpolarization was completely abolished by apamin indicating that it was mediated solely by KCa2.3. (C) Bar graph showing the SLIGRL-induced EDH mediated hyperpolarization (left panel) and relaxation (right panel) in the presence of the NOS inhibitor L-NAME. TRAM-34 significantly attenuated EDH responses and the subsequent addition of t-AUCB had no further effect. Under these conditions, there is no contribution from KCa2.3 channels and tAUCB failed to restore the function of this channel. (D) Bar graphs showing the SLIGRL-induced EDH mediated hyperpolarization (left panel) and relaxation (right panel) in the presence tempol and the NOS inhibitor L-NAME. Tempol restored a KCa2.3 component to the EDH response and also revealed a KCa1.1 component (n = 5). ∗P < 0.05 indicates a significant difference from control, ΨP < 0.05 indicates a significant difference from TRAM-34 + IbTx (KCa2.3 hyperpolarization).

Effect of tAUCB and superoxide scavenging on EDH responses in the presence of L-NAME

In the presence of the NOS inhibitor L-NAME, inhibition of KCa3.1 alone abolishes EDH (McNeish, Dora & Garland, 2005; McNeish et al., 2006). The absence of a KCa2.3 component reflects activation of TP receptors (McNeish & Garland, 2007). Similarly, in this study, the isolated EDH-mediated response obtained in the presence of L-NAME (hyperpolarization of 23.1 ± 4.6 mV and relaxation of 74.3 ± 5.3%, n = 8) was significantly attenuated by TRAM-34 alone (hyperpolarization of −6.5 ± 1.5 mV and relaxation of 25.0 ± 4.2%, n = 5; Fig. 2C). Application of tAUCB to TRAM-34-treated arteries failed to recover the EDH response (hyperpolarization of −4.2 ± 2.4 mV and relaxation of 23.7 ± 2.7%, n = 5; Fig. 2C). In the combined presence of L-NAME and tempol, a cell permeable superoxide scavenger, isolated EDH responses to SLIGRL were significantly inhibited only by the combination of TRAM-34 and apamin. Further inhibition of KCa1.1 with iberiotoxin abolished the residual EDH response (Fig. 2D). Thus, tempol restored the KCa2.3 component to the EDH whereas inhibition of sEH failed to restore this component of EDH.

AA metabolism of the rat middle cerebral artery

Since LOX metabolites mediate a component of EDH, we investigated the metabolism of 14C-AA by cerebral arteries. In arteries treated with the COX inhibitor, indomethacin, to facilitate the identification of LOX metabolites, the products of AA metabolism were resolved by reverse phase HPLC using solvent system 1. 14C-Labelled metabolites co-migrated with the LOX products THETAs, HEETAs, 15-HETE and 12-HETE (Fig. 3A). There were virtually no AA metabolites eluting with the EETs indicating that it is unlikely that the middle cerebral artery produces cytochrome P450 epoxygenase products. In the presence of the NOS inhibitor L-NAME, there was a significant change in the profile of AA metabolism. Metabolites co-migrating with 12-HETE and 15-HETE were markedly increased whereas the metabolites co-migrating with the THETAs and HEETAs increased slightly (Fig. 3B). Arteries were also incubated in the presence and absence of L-NAME and the metabolites resolved by reverse phase HPLC using solvent system 2 that resolves COX metabolites and isoprostanes. Indomethacin was not present in these incubations. Metabolites comigrating with 6-keto PGF1α, 8-iso-PGF2 and 12-HETE were detected (Fig. 4). A 14C-metabolite comigrating with TxB2 was not observed. Following L-NAME treatment, there was also a marked increase in polar metabolites eluting in fractions 5–10 and possibly 8-iso-PGF2, eluting in fractions 33–34 (Fig. 4). Isoprostane synthesis was confirmed by LC/MS analysis. Extracts were purified by reverse phase HPLC using solvent system 3 and the eluate monitored by selected ion monitoring of m/z 351 for PGE2-type isoprostanes and m/z 353 for PGF2-type isoprostanes (Fig. 5). The production of 8-iso-PGE2 was similar in control and L-NAME-treated arteries (380 and 335 ng/ml, respectively; Fig. 5A), whereas 8-iso-PGF2 production increased 3-fold with L-NAME treatment (179 and 583 ng/ml, respectively; Fig. 5B). In addition, several unidentified PGF2-type isoprostanes were produced by cerebral arteries and were increased by L-NAME (Fig. 5B).

Figure 3 Effect of the NOS inhibitor L-NAME on arachidonic acid metabolism by rat cerebral arteries.

Arteries were incubated with 14C-AA and indomethacin in the absence (A) and presence of L-NAME (B). Metabolites were extracted and separated by reverse-phase HPLC, solvent system 1. Migration times of known standards are shown on each chromatogram. CPM, counts per minute; THETA, trihydroxyeicosatrienoic acid; HEETA, hydroxyepoxyeicosatrienoic acid; HETE, hydroxyeicosatetraenoic acid; AA, arachidonic acid.

Figure 4 Effect of L-NAME on arachidonic acid metabolism by rat cerebral arteries.

Arteries were incubated with 14C-labeled arachidonic acid in the absence (control, solid line) and presence of L-NAME (100 mM) (dotted line). Metabolites were extracted and separated by reverse-phase HPLC, solvent system 2. Migration times of known standards are shown on each chromatogram. CPM, counts per minute; PG, prostaglandin; TX, thromboxane; HETE, hydroxyeicosatetraenoic acid; AA, arachidonic acid.

Figure 5 Effect of L-NAME on iso PGE2 and iso PGF2 production of rat cerebral arteries.

Arteries were incubated with AA and indomethacin in the absence and presence of L-NAME. Metabolites were extracted and analyzed by liquid chromatography–mass spectrometry. Selected ion chromatogram m/z 351 (iso PGE2) (A) and m/z 353 (iso PGF2) (B). Migration times of known standards are shown on each chromatogram. * indicates a marked change in production following treatment with L-NAME.

Effect of 8-isoprostane F2 on EDH

Since cerebral arteries synthesize isoprostanes, and L-NAME increased their production, we investigated the effects of 8-iso-PGF2 on vascular tone and EDH. 8-iso-PGF2 caused concentration-dependent contraction of cerebral arteries that was fully inhibited by the TP receptor antagonist SQ 29,548 (10 µM, Fig. 6A).

In control vessels, exogenous application of 8-iso-PGF2 (1 µM) caused significant depolarization and constriction (Em−49.9 ± 2.5 vs −40.7 ± 2.5 mV and tension 1.0 ± 0.2 vs 3.9 ± 0.3 mN pre- and post-8-iso-PGF2, respectively, n = 9, P < 0.05) but did not significantly affect SLIGRL-induced EDH (−28.0 ± 2.7 vs 23.7 ± 2.3 mV, pre- and post-8-iso-PGF2 respectively, n = 9; Figs. 6B and 6C). In the presence of 8-iso-PGF2, EDH was significantly inhibited by TRAM-34 alone (−13.6 ± 2.8 mV, P < 0.05, n = 5; Figs. 6B and 6C) whereas apamin alone had no effect (−25.5 ± 5.9 mV, n = 4; Fig. 6C); nor did it have any additional affect in combination with TRAM-34 (−9.6 ± 1.1 mV, n = 9; Fig. 6C). The residual hyperpolarization was blocked by addition of iberiotoxin (−3.2 ± 0.8 mV, n = 7; Fig. 6C).

Figure 6 Effect of isoprostane F2α (8-iso PGF) on middle cerebral arteries.

(A) Concentration-response curve showing constriction produced by 8-iso PGF in rat middle cerebral arteries that were able to synthesize NO. This constriction was completely blocked by the selective TP antagonist SQ 29,543. (B) Original traces showing the effect of pre-incubation of 8-iso PGF2 on EDH evoked by SLIGRL (20 µM) in rat middle cerebral arteries that were able to synthesize NO. 8-iso PGF2 did not significantly affect control EDH (upper panel) but subsequent inhibition of KCa2.3 (TRAM-34; 1 µM) virtually abolished EDH (lower panel). The vessel viability was assessed by addition of 15 mM K+ which evokes endothelium-independent hyperpolarization and relaxation. (C) Bar graph showing the effect of 8-iso PGF on EDH. Control EDH is unaffected by 8-iso PGF, TRAM-34 alone significantly inhibited EDH and apamin is without effect. The combination of apamin and TRAM-34 had no further effect to TRAM-34 alone. The data shows that 8-iso PGF2 inhibits the KCa2.3 component of EDH normally seen under these conditions. ∗P < 0.05 indicates a significant difference from control. ΨP < 0.05 indicates a significant difference from 8-iso-PGF2α.

Discussion

This study is the first to demonstrate that a LOX metabolite(s) of AA, contributes to endothelium-dependent hyperpolarization (EDH) in the rat middle cerebral artery and thus may be the, as yet, unidentified AA metabolite involved in EDH responses in the middle cerebral artery. Following inhibition of NOS with L-NAME, the profile of AA metabolism is altered. There is no change in production of LOX products; however, L-NAME increased 8-iso-PGF2 production and the production of other iso-PGF2 isomers. We suggest that the increase in isoprostane formation causes the TP receptor-dependent loss of KCa2.3-mediated component of EDH, a phenomenon we had previously attributed to TxA2 (McNeish & Garland, 2007).

The findings that 15-LOX metabolites, namely HEETA and THETA, contribute to the EDH response in rabbit arteries lead us to investigate whether these metabolites contributed to EDH in the rat middle cerebral artery. This was of particular interest as EDH responses in this artery have a component dependent upon an unidentified AA metabolite (McNeish & Garland, 2007; You, Golding & Bryan, 2005). Analysis of AA metabolism by the middle cerebral artery indicates LOX products are generated whereas epoxygenase metabolites such as EETs are not. In control arteries, EDH induced by SLIGRL-NH2 is significantly attenuated by three structurally distinct inhibitors of LOX enzymes. These findings clearly indicate that EDH in this artery is mediated, at least in part, by LOX metabolites.

KCa channels mediate the EDH responses to LOX metabolites; the most likely candidates are either 12-HETE, a 12-LOX metabolite, (Zink et al., 2001) or HEETA and THETA, 15-LOX metabolites (Campbell et al., 2003; Chawengsub et al., 2008; Chawengsub, Gauthier & Campbell, 2009). These metabolites are all produced by the rat middle cerebral artery consistent with the expression of a 12/15-LOX. Interestingly, 12-HETE (30 nM) had no effect on membrane potential or tone in cerebral arteries (data not shown). Therefore, the 15-LOX products, HEETA or THETA, are most likely the AA metabolites that contribute to EDH responses in this artery.

HEETA and THETA stimulate hyperpolarization of rabbit arteries by direct stimulation of KCa2.3 (Campbell et al., 2003; Chawengsub et al., 2008). As HEETA is rapidly metabolized by sEH to THETA, it is difficult to distinguish between the activities of these two eicosanoids. In order to ascertain which of these metabolites contributes to EDH, we used a selective sEH inhibitor tAUCB (Liu et al., 2009); tAUCB does not affect total EDH but selectively augments the KCa2.3 component of EDH. As inhibition of sEH prevents the rapid breakdown of HEETA to THETA, it is likely that the HEETA, not THETA, is the 15-LOX product involved in EDH. In rabbit arteries, both these LOX products directly stimulate KCa2.3 located on the SMC layer and thus act as diffusible EDHFs. However, such an action is unlikely in the rat middle cerebral artery as KCa2.3 are expressed solely on the endothelium (Cipolla et al., 2009; McNeish et al., 2006). Thus, it is more likely that HEETA contribute to EDH by acting as an intracellular or autocrine modulators of endothelial KCa2.3, as proposed for other AA metabolites involved in EDH responses such as EETs (Campbell & Falck, 2007; Edwards, Feletou & Weston, 2010). More recently, metabolites of AA have also been shown to have effects on transient receptor potential channels evoking subsequent KCa channel activation and dilation (Earley, 2011; Sonkusare et al., 2012; Zheng et al., 2013).

Under the experimental conditions described above, the middle cerebral artery produces NO and the KCa2.3 channel contributes to EDH. However, under conditions where NOS is blocked to observe the EDH response in isolation, the KCa2.3 channel component of hyperpolarization is lost and only KCa3.1 channels contribute to the EDH response (Marrelli, Eckmann & Hunte, 2003; McNeish, Dora & Garland, 2005; McNeish et al., 2006). The loss of the KCa2.3 channel component involves stimulation of TP receptors (McNeish & Garland, 2007). Since LOX products appear to selectively augment KCa2.3 channels, we postulated that TP-mediated inhibition of this channel may be due to a reduction in LOX metabolites. However, following NOS inhibition, there was actually an increase in the AA metabolites corresponding to HEETA, THETA and HETEs. Inhibition of sEH (a treatment that would be expected to protect HEETA production and thus KCa2.3 channel activation) failed to restore KCa2.3 channel function when NOS was inhibited. Therefore, loss of KCa2.3 function following NOS inhibition does not involve decreased production of LOX metabolites. This finding supports a previous study that demonstrated inhibition of LOX has no effect on EDH responses following inhibition of NOS (You, Golding & Bryan, 2005).

A potential explanation for the loss of KCa2.3 channel function following NOS inhibition, despite increased production of LOX metabolites, is that TP mediated signaling mechanisms overcome the actions of LOX metabolites. TP receptor stimulation inhibits KCa2.3-mediated responses in control arteries able to synthesize NO (McNeish & Garland, 2007) by a Rho kinase dependent mechanism (McNeish et al., 2012). Therefore, we wished to confirm that synthesis of the endogenous activator of TP receptors, TxA2, is increased following NOS synthase inhibition, as previously suggested (McNeish & Garland, 2007). Surprisingly, we did not detect TxA2 production with, or without, NOS inhibition. However, following NOS inhibition there was an increase in production of isoprostanes which also stimulate TP receptors (Crankshaw, 1995; Elmhurst, Betti & Rangachari, 1997).

The most likely source of isoprostanes is the non-enzymatic reaction of either free or esterified AA with reactive oxygen species such as superoxide (Morrow et al., 1990). Due to our sampling technique of measuring isoprostanes in the incubation medium it is likely that we were measuring isoprostanes produced from free AA (although we cannot eliminate the possibility that esterified isoprostanes were also released into the medium); however the endothelium is the mostly likely source of the isoprostanes as it is the major source of AA metabolites in vascular tissues (Rosolowsky & Campbell, 1993; Rosolowsky & Campbell, 1996). Interestingly, superoxide production is high in the cerebral vasculature when compared to systemic vessels due to increased expression of NADPH oxidase (Miller et al., 2005). As NO reacts rapidly with superoxide, it is conceivable NOS inhibition in the middle cerebral artery increases free superoxide produced from NADPH oxidase and therefore isoprostane synthesis leading to stimulation of TP receptors and loss of KCa2.3 function. The superoxide scavenger tempol prevented loss of KCa2.3 function following inhibition of NOS supporting this hypothesis. Furthermore, using LC/MS, we detected increased production of 8-iso-PGF2 and several as yet unidentified PGF2-type isoprostanes following NOS inhibition. While the source of superoxide was not investigated in this study a similar increase in isoprostane production following inhibition of NOS has been observed kidneys where it was reversed by tempol and knock out of gp91 phox (Haque & Majid, 2008; Kopkan & Majid, 2006), we thus speculate that a similar NADPH dependent mechanism is responsible for increased isoprostane production in cerebral arteries. In arteries not treated with a NOS inhibitor, exogenous 8-iso-PGF2 constricted the middle cerebral artery via a TP receptor dependent mechanism and mimicked the ability of other TP receptor agonists (McNeish & Garland, 2007) to inhibit the KCa2.3 component of EDH. Thus, increased production of 8-iso-PGF2 following NOS inhibition is likely to cause the loss of the KCa2.3 component of EDH.

The data are summarized in Fig. 7, they indicate the novel finding that LOX metabolites of AA, most likely HEETA, contribute to EDH in the middle cerebral artery and that inhibition of sEH with the selective inhibitor tAUCB increases the KCa2.3 component of hyperpolarization. However, the loss of KCa2.3 function following inhibition of NOS is not restored by tAUCB treatment. Instead, NOS inhibition increases F-type isoprostane production resulting in TP receptor stimulation (Fig. 7), the loss of KCa2.3 responses and elimination of the effect of HEETA. These findings may have important implications in diseases associated with loss of NO signaling such as stroke and coronary artery disease where increased isoprostane production has been detected (Patrono & FitzGerald, 1997). This pathway may be related to the severity of the disease (Wang et al., 2006). Our data indicate that inhibition of sEH may not be an effective treatment for cerebrovascular disease but prevention of isoprostane formation may be beneficial.

Figure 7 Schematic showing the role of LOX products on EDH responses in the middle cerebral artery, the proposed effect that inhibition of NOS and increase in isoprostane production has on the EDH response.

(1) Under conditions where middle cerebral arteries can synthesize NO, metabolism of AA by LOX produces HEETA which is metabolized by sEH to THETA. Both can activate KCa2.3 channels. As tAUCB augments the KCa2.3 response HEETA is likely to be the metabolite involved. (2) Under these conditions the EDH response also involves activation of KCa3.1 and KCa1.1 channels to produce smooth muscle cell hyperpolarization and relaxation. (3) When NO is synthesized (4) it rapidly reacts with superoxide preventing production of isoprostanes, when NOS is inhibited (5) this inhibitory effect of NO on isoprostane production is reduced. Endothelial cells are the major source of vascular AA metabolites but it is possible NO prevents isoprostane production in SMCs. The subsequent stimulation of the TP receptor (6) inhibits the KCa2.3 component of EDH as we have previously reported (McNeish & Garland, 2007; McNeish et al., 2012). Solid lines/arrows represent known pathways, dashed lines indicate possible/unconfirmed pathways, green arrows are stimulatory pathways, red lines/text represent inhibitory pathways/blocking agents. EC, endothelial cell; SMC, smooth muscle cell.

Abbreviations

12-HETE 12-hydroxyeicosatetraenoic acid

AA arachidonic acid

COX cyclooxygenase

CYP450 cytochrome P450 epoxygenases

EDH endothelium dependent hyperpolarization

EDHF endothelium derived hyperpolarizing factor

EETs epoxyeicosatrienoic acids

HEETA 15-hydroxy-11,12-epoxyeicosatrienoic acid

KCa1.1 large conductance calcium-activated potassium channel (BKCa)

KCa2.3 small conductance calcium-activated potassium channel (SKCa)

KCa3.1 intermediate conductance calcium-activated potassium channel (IKCa)

LOX lipoxygenase

NOS NO synthase

sEH soluble epoxide hydrolase

SLIGRL serine, leucine, isoleucine, glycine, arginine, leucine

THETA 11,12,15-trihydroxyeicosatrienoic acid

TxA2 thromboxane A2

TP thromboxane prostanoid receptor

tAUCB was a generous gift from Professor Bruce D. Hammock (University of California, Davis).

Additional Information and Declarations

Competing Interests

Author Contributions

Animal Ethics

The authors declare there are no competing interests.

Kathryn M. Gauthier conceived and designed the experiments, performed the experiments, analyzed the data, contributed reagents/materials/analysis tools, prepared figures and/or tables.

William B. Campbell conceived and designed the experiments, contributed reagents/materials/analysis tools, wrote the paper, prepared figures and/or tables, reviewed drafts of the paper.

Alister J. McNeish conceived and designed the experiments, performed the experiments, analyzed the data, contributed reagents/materials/analysis tools, wrote the paper, prepared figures and/or tables, reviewed drafts of the paper.

The following information was supplied relating to ethical approvals (i.e., approving body and any reference numbers):

All animal work in the UK was carried out in accordance to schedule 1 of the Animals (scientific procedures) Act 1986. The establishment licence number registered with the UK Home Office is 30/2307. In the USA it was conducted according to the Medical College of Wisconsin Institutional Animal Care and Use Committee under Animal Use Applications AUA00000041 and AUA00000899.

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
