# Peer review of "Regulation of KCa2.3 and endothelium-dependent hyperpolarization (EDH) in the rat middle cerebral artery: the role of lipoxygenase metabolites and isoprostanes"

_PeerJ, doi:10.7717/peerj.414_

## Round 0.1 · original submission · Minor Revisions

I agree with the reviewers. Please, address the points raised by the two reviewers that recommended publication after revision.

Reviewer 1 ·

Basic reporting

This is a well written article investigating the identity of an endothelium derived hyperpolarising factor in the rat middle cerebral artery.

Figure legends do not match figures, the labelling has become confused.

Experimental design

The methods are comprehensively detailed. Although commonly practised, why is the brain and middle cerebral artery placed in ice cold krebs?

Minor point
It is the 1986 not 1988 Animals (scientific procedures) act.

Validity of the findings

The results suggest that a LOX metabolite is involved in the EDH response in the rat middle cerebral artery.

In figure 1 the authors show that NDGA and ebselen both reduce the EDH response which is further attenuated by TRAM-34. They rightly describe that addition of apamin and iberiotoxin does not further inhibit the EDH response with NDGA, however they suggest the same with ebselen, which does not appear to the case. The addition of apamin and iberiotoxin does appear to further inhibit the EDH response. The authors state that one way ANOVA followed by Tukeys post test has been used, but there is no reference to statistical significances between the different treatment groups e.g. ebselen and ebselen + TRAM-34. If the authors wish to only show statistical difference between control and all treatment groups then a one way ANOVA followed by a Dunnets post test would be more appropriate.

Line 172/173 - the authors state that TRAM-34 or apamin alone do not cause significant inhibition (figure 1), however only the data for TRAM-34 is shown.

Additional comments

Overall this is concise article describing a potential role for LOX metabolites that has been conducted to a good standard.

·

Basic reporting

No comments

Experimental design

No comments

Validity of the findings

No comments

Additional comments

This study investigates the role of 12/15LOX metabolites in the EDH response in the rat middle cerebral arteries. The techniques, microelectrode measurements of membrane potential and measurements of MCA relaxation to the PAR 2 agonist SLIGR, and a range of selective pharmacological inhibitors chosen to dissect individual KCa components contributing to EDH in this preparation were adequate and appropriate for the purpose of this study. A number of novel and interesting findings were reported. Using three different LOX inhibitors, it was demonstrated that EDH response due to activation of KCa2.3 and KCa3.1 was reduced suggesting the involvement of LOX metabolites in this preparation. A selective inhibitor of soluble epoxide hydrolase (an enzyme metabolising HEETAs to THETAs,) tAUCB selectively enhanced
the KCa2.3 component of EDH in preparations with active NOS, implying that HEETAs could be and an actvator of KCa2.3. This enhancement was not observed when after the inhibition of NOS with L-NAME, but was restored in the presence of the superoxide scavenger tempol, suggesting that ROS may be involved. The nature of 15- and 12-LOX products under control conditions was evaluated using HPLC and 14C-AA as precursor. In the presence of L-NAME very little change in LOX metabolites was seen, instead F-type isoprostanes were increased. This consequently led to the assessment of the effect of 8-iso-PGF2alpha on Em and relaxation and providing the evidence that the KCa2.3 component of EDH could be blocked by activation of TP receptors (that normally could be activated by TxA2) by increased production of F-isoprostanes in the MCA. Overall, it is well planned and well performed study, although the MS requires a careful proofreading to make improve its clarity

There are few questions I would like the authors to discuss.

From discussion the relation between NOS inhibition and superoxide production and isoprostanes production is not very clear. It is suggested that NO rapidly scavenging superoxide and when NOS is blocked superoxide is increased isoprostane synthesis is increased as a result (lines 321-323). As it has been mentioned above (lines 320-321) the basal levels are greater in MAC, but what could be a potential source of superoxide in a non-stimulated MCA? Considering the approach used in this study, could it be a shift in the balance between NOS functioning as the NO producer and as ‘NOX’ producing superoxide (another known function of NOS)? What could be the source of F-isoprostanes, free AA or that which is still in membrane bound phospholipids? One would expect that free AA will be randomly oxidized by free radicals, yet the results shown in Fig.5 suggest a targeted increase in 8-iso-PGF2alpha? It might be worth thinking about drawing a simple diagram showing possible pathways and mediators responsible for the EDH response in this preparation. That should increase further the impact of the paper.

What would be a possible mechanism of inhibition of KCa2.3 mediated by activation of TP receptors? It is unlikely to be an increase in Ca concentration that these receptors are capable to produce.

HPLC studies were performed on preparations containing indomethacin. This has certain implications on the interpretation of the results. Firstly, this would explain why no changes TxA2 where measured (lines 316-317). Also, inhibition of COX would shift the balance between AA metabolites may have implication on the estimation of relative contribution of LOX metabolites and isoprostanes produced in conditions when COX is inhibited compared to those when it is intact. These shortcomings need to be properly addressed in discussion.


Minor queries.

What concentration of tempol was used? Please indicate in methods.

L174, L179 L186 & others “-“sign is missing by EDH in mV.

When describe the multi-panel figures in the text, please indicate which figure panel you describe (e.g. 2B , 2C etc), it is only done in some cases.

In figure legends, referring to bar graphs as histograms is not accurate. Please re-phrase.

---

## Round 0.2 · accepted · Accept

Thank you for the changes in the revised manuscript that properly answered the reviewers.